# The Ground-Based Absolute Radiometric Calibration of the Landsat 9 Operational Land Imager

Jeffrey S. Czapla-Myers [1,*], Kurtis J. Thome [2], Nikolaus J. Anderson [1], Larry M. Leigh [3], Cibele Teixeira Pinto [3] and Brian N. Wenny [4]

1 Remote Sensing Group, Wyant College of Optical Sciences, University of Arizona, Tucson, AZ 85721, USA; nanderson@optics.arizona.edu
2 NASA Goddard Space Flight Center, Code 618, Greenbelt, MD 20771, USA; kurtis.thome@nasa.gov
3 Office of Engineering Research, College of Engineering, South Dakota State University, Brookings, SD 57007, USA; larry.leigh@sdstate.edu (L.M.L.); cibele.teixeirapinto@sdstate.edu (C.T.P.)
4 Science Systems & Applications Inc., Lanham, MD 20706, USA; brian.n.wenny@nasa.gov
* Correspondence: jscm@optics.arizona.edu; Tel.: +1-520-621-4242

**Abstract:** This paper presents the initial vicarious radiometric calibration results for Landsat 9 OLI using a combination of ground-based techniques and test sites located in Nevada, California, and South Dakota, USA. The field data collection methods include the traditional reflectance-based approach and the automated Radiometric Calibration Test Site (RadCaTS). The results for top-of-atmosphere spectral radiance show an average ratio (OLI/ground measurements) of 1.03, 1.01, 1.00, 1.02, 1.02, 1.01, 0.98, and 1.01 for Landsat 9 OLI bands 1–8, which is within the design specification of ±5% for spectral radiance. The results for top-of-atmosphere reflectance show an average ratio (OLI/ground measurements) of 0.99, 0.99, 1.00, 1.02, 1.01, 1.02, 1.00, and 1.00 for Landsat 9 OLI bands 1–8, which is within the design specification of ±3% for top-of-atmosphere reflectance.

**Keywords:** Landsat 9; OLI; radiometric calibration; absolute calibration; radiometric validation

## 1. Introduction

The Landsat program is the longest-running Earth observation satellite program in history, providing a continuous record of the Earth's land surface since the launch of the first Landsat satellite, Landsat 1, in 1972. The program is a joint effort between NASA and the United States Geological Survey (USGS) and currently consists of three operational satellites as follows: Landsat 7, Landsat 8, and Landsat 9. The Landsat program has had a significant impact on various fields, including agriculture, forestry, geology, and land use planning. It has been used to monitor natural disasters, track urban growth, and map global land cover changes. In addition, the program has facilitated the development of numerous applications and technologies, including Geographic Information Systems (GISs), remote sensing, and image processing techniques.

Over the years, the Landsat satellites have undergone several improvements in terms of technology, data quality, and coverage. The Landsat 1–3 satellites were equipped with a Multispectral Scanner (MSS) that had four spectral bands. Landsat 4 and 5 carried the Thematic Mapper (TM) instrument, which provided more detailed and accurate data with seven spectral bands. Landsat 7 launched in 1999 with the Enhanced Thematic Mapper Plus (ETM+) instrument, which added a panchromatic band, improved geolocation accuracy, and onboard calibration. Today, the Landsat program continues to provide valuable data for scientific research, resource management, and decision making at local, national, and global scales. Its legacy is a testament to the importance of long-term Earth observation programs for understanding and managing the planet.

The most recent additions to the program are Landsat 8 and Landsat 9, launched in 2013 and 2021, respectively. They carry essentially identical payloads, which include

advanced multispectral sensors that operate in the visible and near-infrared (VNIR), short-wave infrared (SWIR), and thermal infrared (TIR) spectral regions. Additional upgrades include improved spatial resolution and data quality. Landsat 9 is the most recent satellite in the continuous fifty year history of the Landsat program, which will continue with the Landsat Next program [1,2]. It was launched on 27 September 2021 from the Vandenberg Space Force Base using a United Launch Alliance Atlas V 401 launch vehicle and placed into a sun-synchronous 705 km orbit with a sixteen day repeat cycle. It is effectively in the same equatorial crossing time as Landsat 7 before the lowering of its orbit, which is part of its decommissioning phase.

The Landsat 9 payload consists of the same two instruments on Landsat 8 as follows: the Operational Land Imager (OLI) and the Thermal Infrared Sensor (TIRS), which provide complimentary spectral coverage. The OLI operates in the solar-reflective spectral region (400 nm to 2500 nm) using nine spectral channels, while the TIRS is a two-band thermal sensor that operates from 10.6 μm to 12.5 μm. The Landsat 9 OLI spectral bands retain the same spatial characteristics as those of Landsat 8 OLI and Landsat 7 ETM+, which is a 30 m ground sampling distance (GSD) for the multispectral bands, a 15 m GSD for the panchromatic band, and a 185 km swath width (Table 1). The most notable change in recent Landsat solar-reflective bands is the transition from a whiskbroom sensor to a pushbroom sensor (ETM+ vs. OLI), which allows for a higher signal-to-noise ratio (SNR). ETM+ has an 8-bit radiometric resolution, while the Landsat 8 OLI has a 12-bit radiometric resolution. An additional improvement for the Landsat 9 OLI is that of a 14-bit radiometric resolution, which allows for a higher detection change over dark targets such as vegetation and water [3,4]. The relative spectral responses (RSRs) for the OLI multispectral bands 1–7 are shown in Figure 1.

**Table 1.** The spectral bands and GSD of the Landsat 9 OLI. The center wavelength defined here is the band average of the relative spectral response (RSR) for each band.

| Band | Center Wavelength (nm) | Bandwidth (FWHM, nm) | GSD (m) |
|---|---|---|---|
| | | **Landsat 9 OLI** | |
| 1 | 443 | 16 | 30 |
| 2 | 483 | 60 | 30 |
| 3 | 561 | 57 | 30 |
| 4 | 655 | 37 | 30 |
| 5 | 865 | 29 | 30 |
| 6 | 1609 | 86 | 30 |
| 7 | 2201 | 189 | 30 |
| 8 (pan) | 592 | 172 | 15 |

The requirement for the absolute radiometric uncertainty of the Landsat 9 OLI is the same as that for the Landsat 8 OLI as follows: ±5% for top-of-atmosphere (TOA) spectral radiance and ±3% for TOA reflectance (both at the k = 1 level). The Landsat 9 OLI onboard radiometric calibration system is identical to that of Landsat 8 OLI, and the various subsystems include a shutter for dark measurements, two solar diffusers (working and pristine), two stimulation lamp assemblies, and the ability to view the moon through a spacecraft maneuver [2,5]. As with the Landsat 8 OLI, the Landsat 9 OLI underwent a rigorous pre-launch calibration and characterization process [6–10].

The radiometric calibration and validation of Earth-observing sensors continues to be an essential component of post-launch operation. Monitoring the spectral, radiometric, and geometric temporal changes in a sensor is critical for the scientific community, who rely on these data for a wide variety of scientific, environmental, and societal applications. Ground-based techniques are now in a mature state and continue to be used to monitor and characterize any changes that may occur during the operational life of a sensor. The value of these systems is their independence from any onboard calibration systems that may also change over time. Independent radiometric validation has also become more important

with the growing number of operational satellite constellations, many of which consist of CubeSats or nanosats that typically lack onboard calibration systems. Current ground-based techniques include automated instrumented sites [11], in situ measurements by on-site personnel [12–18], pseudo-invariant calibration sites (PICSs) [19], lunar observations, and mirror-based artificial targets [20,21]. Cross-calibration with suitable spaceborne and airborne sensors is also widely used for calibration and validation studies. The independent validation of Earth-observing systems ensures that data from sensors with different spectral, spatial, and temporal characteristics can be placed on the same traceable radiometric scale, thereby allowing for harmonized data products with a well-defined uncertainty.

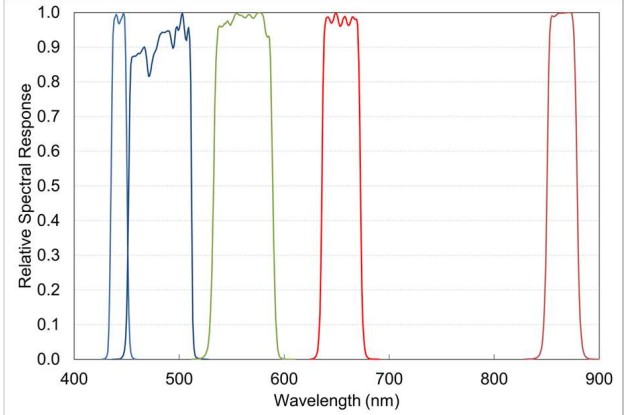 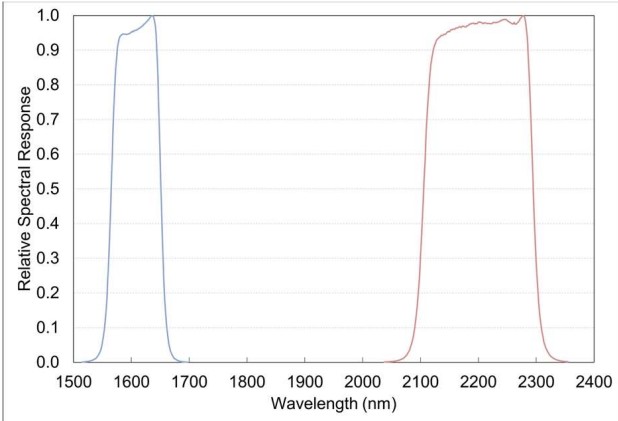

**Figure 1.** The relative spectral response (RSR) for the Landsat 9 OLI multispectral bands 1–5 (**left**) as well as 6 and 7 (**right**). The panchromatic band 8 is omitted for clarity, and the cirrus band 9 is also excluded since it is not used in this work.

Vicarious radiometric calibration is a ground-based technique that has been in use for almost 40 years [22–24]. It provides an independent analysis that measures and monitors any post-launch changes in the airborne or spaceborne system under test. This work presents the results from the initial nineteen months after the launch of the Landsat 9 OLI and includes measurements using on-site personnel as well as those using an instrumented automated test site. The Landsat 9 OLI results from the tandem flight phase of Landsat 8 and 9 are also presented. Section 2 of this paper presents the reflectance-based and RadCaTS ground-based methodologies. Section 3 describes the field campaigns during the nineteen-month period of this work (November 2021 to June 2023), and Section 4 summarizes the results for each of the ground collects. Section 5 presents a brief description of the ground measurement uncertainty, and Section 6 provides a summary of the results and conclusions of this work.

## 2. Methodology

### 2.1. Ground-Based Vicarious Radiometric Calibration

In general terms, ground-based vicarious radiometric calibration uses scenes of known spectral radiance and compares them to a sensor under a test to evaluate its absolute radiometric calibration. In the solar-reflective regime, the creation of a scene with known spectral radiance requires knowledge of the surface reflectance [25] and the atmospheric transmittance [18], as well as knowledge of the atmospheric scattering properties. The surface reflectance is typically measured by ground instrumentation that may or may not require on-site personnel. Atmospheric measurements are made using suitable spectrora-diometers, with two examples being the Automated Solar Radiometer (ASR) [26–28] and the Cimel CE-318 solar–lunar photometer [29,30]. In general terms, the results of these two measurements are used in a radiative transfer code to determine the at-sensor spectral radiance for a given airborne or spaceborne instrument [31]. The work described in this paper uses two independent ground-based techniques, the first of which uses a system of auto-

mated instruments to make surface reflectance and atmospheric measurements (RadCaTS), while the other uses ground-based personnel for similar measurements (reflectance-based approach). Each of these systems is described in the next two sections, with the RadCaTS being introduced first since it is now the primary data collection technique used by the University of Arizona (UArizona).

*2.2. Automated Measurements: The Radiometric Calibration Test Site (RadCaTS)*

2.2.1. RadCaTS Development

The RadCaTS was developed by the Remote Sensing Group of the Wyant College of Optical Sciences at the University of Arizona (UArizona) in the early 2000s [32–34]. During the prototyping phase, the RadCaTS was used to supplement the in situ data that were routinely collected by on-site personnel using the traditional reflectance-based approach. A data processing methodology was developed, tested, and compared to the reflectance-based results during the prototyping stage. The knowledge gained in this process resulted in the development of radiometrically stable, all-weather, multispectral ground-viewing radiometers (GVRs), the first examples of which were deployed in 2012 [35,36]. Additional upgrades over the past ten years have included a satellite uplink station, upgraded Cimel CE-318T solar–lunar photometer [29], and a GVR with linear motion [37]. The RadCaTS has been used successfully for such sensors as the Landsat 7 ETM+ [38], the Landsat 8 and 9 OLIs [17,39,40], Terra and Aqua MODIS [41], ASTER [42,43], SNPP, NOAA-20 VIIRS [44], Sentinel-2A and -2B MSI [38], and GOES-16 and -17 ABI [45,46]. It has also been used to validate the calibration of airborne sensors [15,47,48].

The expansion from on-site personnel to an automated system evolved as a sensible response to the growing number of national and international Earth-observing satellite programs. In addition to national space programs, the creation of nanosat and CubeSat constellations by commercial companies is also increasing rapidly. Automated ground-based radiometric calibration sources become increasingly important to the operators of these programs since many of these small satellites have limited or no onboard radiometric calibration systems [49–58].

2.2.2. RadCaTS: Part of a Global Radiometric Calibration Network

The RadCaTS is one of the four original sites in the Radiometric Calibration Network Working Group (RadCalNet), which was established by the Committee on Earth Observing Satellites (CEOS) Working Group on Calibration and Validation (WGCV) Infrared and Visible Optical Sensors (IVOS) [11]. There are currently five instrumented sites located in the USA, France, Namibia, and China, with new sites currently under construction and evaluation [59]. TOA reflectance, bottom-of-atmosphere (BOA) reflectance (or the surface BRF, as denoted in this work), and various atmospheric parameters from all sites are all freely available to registered users at the RadCalNet web portal (www.radcalnet.org, accessed on 1 January 2024). RadCalNet data are currently being used to validate the calibration of a variety of spaceborne and airborne sensors [15,47,60–70]. The web portal also contains documentation for those interested in developing a RadCalNet site, information for the users of RadCalNet data, and historical intercomparison reports from a RadCalNet user workshop. Additionally, the portal hosts the RadCalNet Forum, which allows users to interact directly with the RadCalNet Working Group, including site owners and data providers (forum.radcalnet.org, accessed on 1 January 2024). Notices regarding site status, data availability, and data quality are provided by site operators on a regular basis in order to keep users informed of data outages or quality issues.

2.2.3. Development of Custom Ground-Viewing Radiometers for RadCaTS

In 2002, the RadCaTS concept started with a one-channel proof-of-concept radiometer that used a light-emitting diode (LED) as a detector. The decision to use an LED as a detector was influenced by the successful Global Learning and Observations to Benefit the Environment (GLOBE) program, which promotes the collection and analysis of environmental data

as well as the collaboration of students around the world [71–74]. The preliminary RadCaTS radiometer was based on the GLOBE sun photometer, and prototypes were constructed, characterized, and field-tested using various VNIR LEDs. Modest improvements to the radiometer design over the next few years included the addition of two more channels in the VNIR region, as well as powered optics to better control the field of view [75]. The proof-of-concept instruments allowed the data processing architecture to be developed and tested, and it was determined that to be a truly operational and well-calibrated radiometric resource for the Earth observation community, new radiometers had to be developed.

In 2012, the prototype three-channel VNIR GVRs were replaced with a new version developed by UArizona to transition to a truly operational and autonomous test site. Improvements include a temperature-controlled focal plane, interference filters for spectral selection, and a solar power supply for remote operation. The eight spectral channels of the GVRs used at the RadCaTS are nominally centered at 400 nm, 450 nm, 500 nm, 550 nm, 650 nm, 850 nm, 1000 nm, and 1550 nm, and they have 20 nm bandwidths (full width at half maximum, FWHM). The focal plane of each GVR is mounted 1.5 m above the ground, which corresponds to a spot-size diameter on the ground of 27 cm, as defined by the 10° full field of view for each channel. The GVRs were designed, developed, tested, and characterized at UArizona specifically for the spectral and spatial characteristics of the surface at Railroad Valley.

In the original operational phase in 2012, there were three GVRs deployed to the RadCaTS, and in 2014, a fourth one was deployed. The development of more GVRs continued throughout the years, along with the addition of a satellite communication network that allows data to be uploaded daily to UArizona for processing. There are currently seven GVRs at the RadCaTS in a nadir-viewing configuration. In the past, two GVRs were mounted in the viewing configuration of the GOES-East and GOES-West geostationary satellites for work with the Advanced Baseline Imager (ABI), but they were reverted to nadir-viewing in 2020 [45,46]. The positions and number of the GVRs required to obtain the same uncertainty in the surface bidirectional reflectance factor (BRF) as the reflectance-based approach were assessed using high-resolution satellite imagery [33,75–77]. Examples of the current GVRs at the RadCaTS are shown in Figure 2, which also shows a recent prototype design that uses a new mount for the radiometer head. The new mount allows for 90 cm of linear motion, which increases the spatial sampling. The current positions of the seven GVRs are shown in Figure 3.

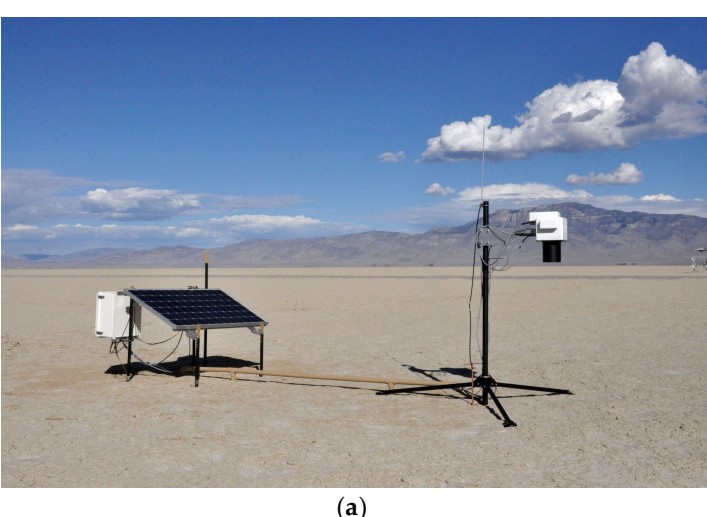 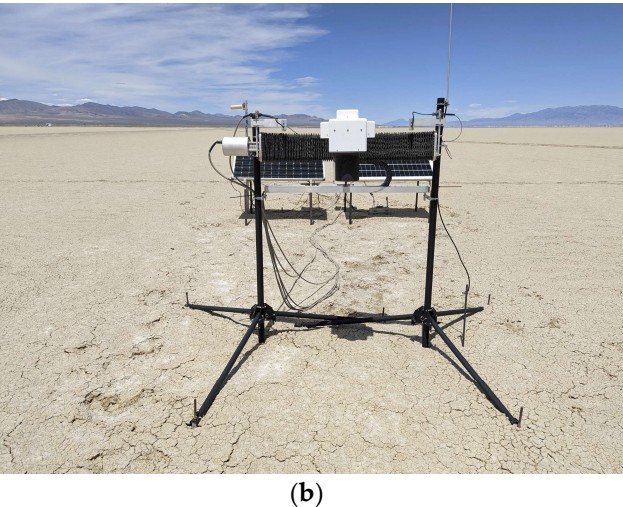

(**a**)  (**b**)

**Figure 2.** (**a**) An example of two GVRs at the RadCaTS. Currently, six of the seven have a static mount. (**b**) A GVR with a linear translation mount that provides 90 cm of motion.

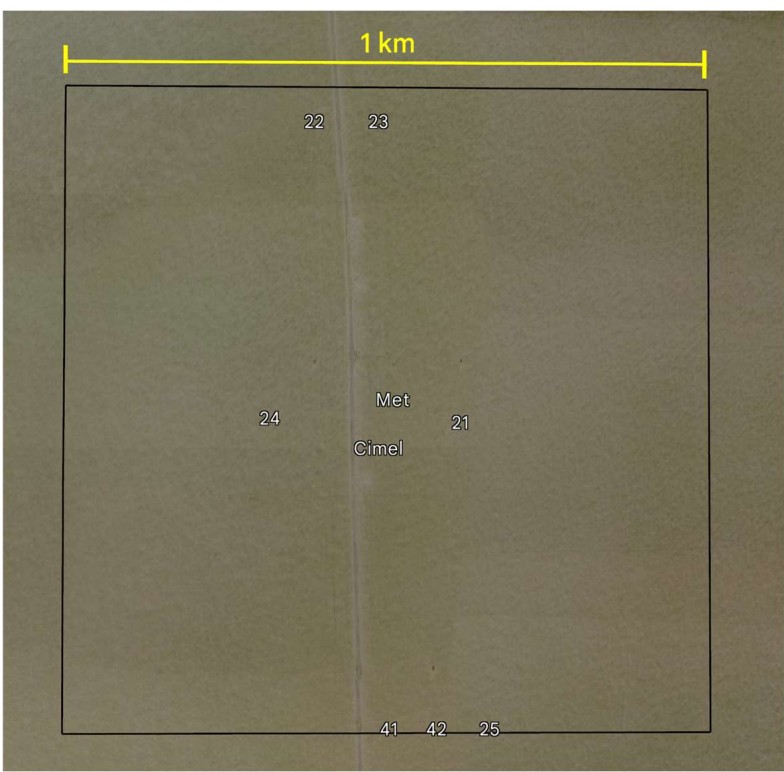

**Figure 3.** The current locations of the seven RadCaTS GVRs (21–25, 41, and 42), the meteorological station (Met), and Cimel sun photometer(s) (Cimel) at Railroad Valley. GVR 23 has 90 cm of linear motion. The feature running north–south is a gravel road.

### 2.2.4. Railroad Valley, Nevada, USA

The RadCaTS is located at Railroad Valley, Nevada, USA, which has a usable area of 15 km × 15 km at an altitude of 1435 m. Preliminary work by UArizona in the 1990s sought to identify suitable ground test sites in the Southwestern United States for the reflectance-based approach and also for the cross calibration of Earth-observing sensors that operate in the solar-reflective regime (400 nm to 2500 nm) [78]. Examples of the principal criteria for a suitable site included the following:

- Surface reflectance: >0.3, with the aim to reduce uncertainties in the path radiance.
- Spatial uniformity: to reduce uncertainties due to sensor misregistration during cross calibration studies.
- Large size: to reduce uncertainties from adjacency effects.
- Arid region: to reduce surface reflectance changes due to precipitation and/or the presence of clouds.
- High altitude: to reduce uncertainties in atmospheric characterization due to aerosols.
- Accessibility: in the 1990s, UArizona deployed a mobile lab pulled by a truck during field campaigns, so the test site had to be accessible with the truck and trailer.

A preliminary exploratory trip to Railroad Valley occurred in 1996, and regular field work commenced soon after. Other sites used by UArizona include Lunar Lake, Nevada; [79,80], Ivanpah Playa, California; and White Sands Missile Range, New Mexico [81]. By early 2000, Railroad Valley and Ivanpah Playa were the two main sites used by the group for sensors such as Landsat 5 TM, Landsat 7 ETM+, Aqua and Terra MODIS, and ASTER. There were three main collection areas used by UArizona at Railroad Valley as follows: one for pushbroom sensors (e.g., ASTER), one for whiskbroom sensors (e.g., TM and ETM+), and one for large-footprint sensors such as MODIS. The original concept for the RadCaTS design specified that it was to be suitable for moderate-resolution sensors such as MODIS, so the large-footprint area at Railroad Valley was chosen as the region of interest (ROI) for

the RadCaTS. The choice to use a preexisting area was also important because it meant that a library of surface BRF measurements was already being established.

The RadCaTS ROI is 1 km $\times$ 1 km in size and centered at 38.497° latitude, $-115.690°$ longitude. In terms of the World Reference System-2 (WRS-2), which is a global notation system for Landsat data, it is path 40 and row 33. Figure 4 shows a Landsat 9 OLI image of Railroad Valley collected on 10 July 2022 at 18:21 UTC. The 1 km$^2$ RadCaTS ROI shown in Figure 3 is indicated by the yellow box in Figure 4.

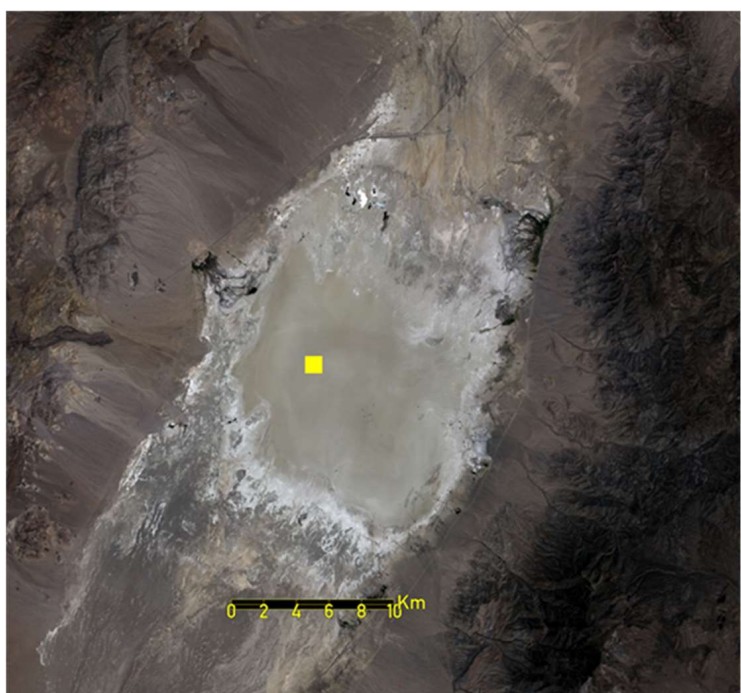

**Figure 4.** A Landsat 9 OLI image of Railroad Valley (10 July 2022, 18:21 UTC) created using OLI bands 4, 3, and 2. The yellow box denotes the 1 km $\times$ 1 km RadCaTS ROI (WRS-2 path 40, row 33).

### 2.2.5. Atmospheric Measurements at the RadCaTS

The atmospheric measurements at the RadCaTS are performed using a Cimel sun photometer that follows the AERONET measurment collection protocol [29,30,82–84]. Throughout the day, direct solar irradiance and sky radiance measurements are made automatically, and the data are uploaded hourly through a satellite uplink to the NASA Goddard Space Flight Center (GSFC) for further processing, where they are subsequently made available for public access on the AERONET web portal. In the case of the RadCaTS, the atmospheric quantities required as the input to the radiative transfer code include the following:

- Aerosol optical depth (AOD);
- Angstrom exponent;
- Columnar water vapor;
- Columnar ozone;
- Carbon dioxide.

The AOD, Angstrom exponent, and columnar water vapor data are obtained from AERONET, while the columnar ozone data are obtained from either the Ozone Monitoring Instrument (OMI) or the Ozone Mapping and Profiler Suite (OMPS) data portals. The daily carbon dioxide amount is downloaded from the NOAA Global Monitoring Laboratory (GML) web portal (gml.noaa.gov, accessed on 1 February 2024).

Once these data have been acquired, they are used as input into the radiative transfer code to determine the spectral transmission of the direct solar beam, as well as the diffuse sky irradiance. The MODTRAN mid-latitude summer atmospheric profile and

corresponding standard atmosphere vertical profile column amounts are used for the Rad-CaTS. The exoatmospheric solar irradiance model used for Landsat work is Chance–Kurucz (ChKur) [85]. The aerosol optical properties are determined using an internal database of Mie-generated aerosol phase functions in MODTRAN. The aerosol optical properties at the RadCaTS are assumed to follow a power-law size distribution that has a given Angstrom exponent and complex refractive index. The use of such an assumption at desert sites such as Railroad Valley has been justified because of the typical low amount of aerosol loading coupled with a relatively high surface BRF. The MODTRAN output of interest for the RadCaTS is the atmospheric spectral transmittance from 350 nm to 2500 nm as well as the diffuse sky irradiance, $E_{sky}$, which is used in the determination of the surface BRF, described in the following section.

### 2.2.6. Surface Reflectance Measurements at the RadCaTS

One of the primary differences between the RadCaTS and the traditional reflectance-based approach is that of the methodology used to determine the surface BRF. The RadCaTS uses GVRs to measure reflected radiance from the test site surface, while the reflectance-based approach uses the ratio of digital numbers (DNs) collected with a portable spectrora-diometer over the ground to those collected over a reference panel [18,42,86–91]. The most notable difference is that the RadCaTS requires a solar exoatmospheric irradiance model to convert the measured reflected surface radiance to a surface BRF [17].

Operationally, the BRF of the RadCaTS ROI is determined for a given time of interest by initially determining the BRF in all eight channels of each GVR. This step requires the spectral radiance measurement in each GVR channel, the atmospheric measurements made by the Cimel sun photometers, the ancillary data (e.g., ambient temperature and pressure), and an exoatmospheric irradiance model. These data are processed in the radiative transfer model (currently MODTRAN 6 [31]), and the output is the multispectral surface BRF, an example of which is shown in Figure 5 for a Landsat 9 overpass on 24 June 2022.

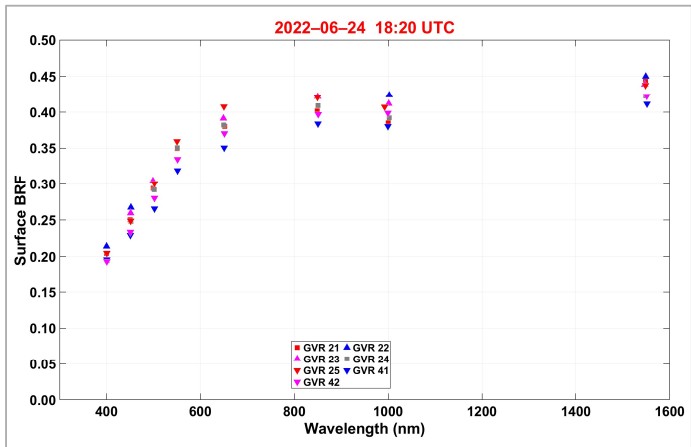

**Figure 5.** The retrieved multispectral BRF in each of the eight channels of seven GVRs on 24 June 2022 at 18:20 UTC. Their geolocation at the RadCaTS is shown in Figure 3.

The final step in processing the surface reflectance measurements is to convert the multispectral surface BRF to a hyperspectral BRF for use once again in the radiative transfer code. The conversion is completed using a least-squares fit of the multispectral GVR data to one of the hyperspectral data sets in a reference library, which is a collection of portable spectroradiometer measurements made by UArizona personnel over the past thirty years. UArizona personnel are the main source for these data, but more recent BRF data have been obtained from NASA JPL personnel who perform similar measurements for different sensors [48,92,93]. A limited amount of snow BRF data have been collected by UArizona personnel during various field campaigns at Railroad Valley, but they are generally not used in RadCaTS or RadCalNet processing. An example of the resulting conversion from

multispectral to hyperspectral BRF is shown in Figure 6. The procedure to determine the surface BRF for a given date and time is summarized below and also in Figure 7. The equation for the surface BRF determination is shown in Appendix A.

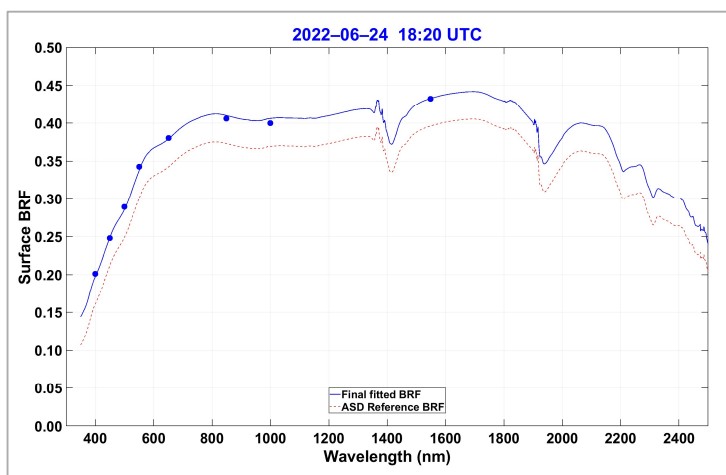

**Figure 6.** The average multispectral BRF retrieved from the seven GVRs in Figure 5, including the fit to one of the hyperspectral portable spectroradiometer data in the reference library.

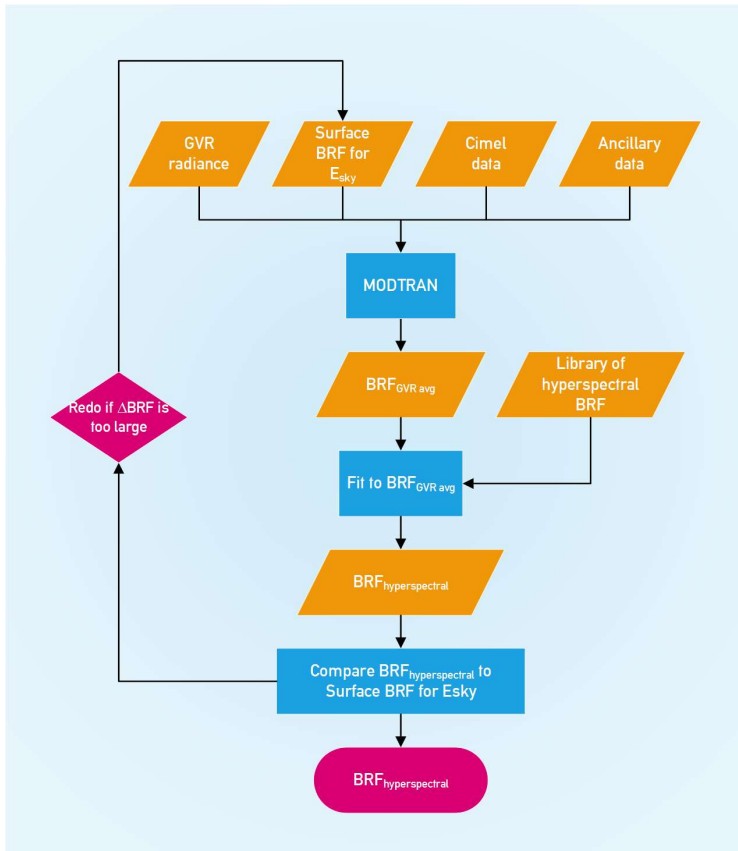

**Figure 7.** RadCaTS surface BRF processing.

1. Determine the surface BRF in each GVR channel.
   - Calculate the spectral radiance measured by each GVR channel.
   - Use a reference monthly average BRF for the diffuse sky irradiance ($E_{sky}$) calculation.

- Obtain the processed AERONET data for the time of interest, including the $AOD_{500nm}$, precipitable water vapor (WV) and Angstrom exponent.
- Download atmospheric data such as ozone and $CO_2$ amount.
- Download ancillary data such as ambient temperature and barometric pressure from the on-site meteorological station at Railroad Valley.
- Use the AERONET measurements, the atmospheric data ($CO_2$ and $O_3$), and ambient temperature and pressure data as input into a radiative transfer code.

2. Convert the multispectral GVR surface BRF to a hyperspectral BRF.
   - Compute the average surface BRF for each of the eight GVR bands in order to obtain one multispectral surface BRF for the RadCaTS ROI.
   - Perform a least-squares best fit of the multispectral surface BRF to a library of reference BRF values obtained with multispectral spectroradiometers.

3. Compare the output hyperspectral surface BRF with the one used in 1b.
   - If the average difference is higher than a predetermined value, rerun step 1 using the new hyperspectral surface BRF.
   - Continue this process until the difference between the two values converge to being within the predetermined value. (Note: the wavelength regions used for this comparison are as follows: 400 nm to 1200 nm, 1500 nm to 1700 nm, and 2000 nm to 2250 nm. These spectral regions are chosen in order to avoid absorption regions in the atmosphere.)

4. At this point, the hyperspectral surface BRF has been determined for the given time and date of interest.

### 2.2.7. Determination of TOA Spectral Radiance and TOA Reflectance

Once the hyperspectral surface BRF has been determined at the RadCaTS for the date and time of interest, the TOA spectral radiance and reflectance are determined using one final run of the radiative transfer code. The resulting output is the hyperspectral TOA spectral radiance (and reflectance, if applicable), which is then band-averaged to the bands of the sensor under testing. In the final step, these values are compared to the sensor imagery, which is downloaded from the USGS Earth Explorer web portal in the case of Landsat. An example of the hyperspectral TOA spectral radiance determined at the RadCaTS for the Landsat 9 overpass on 24 June 2022 is shown in Figure 8, and a summary of the processing scheme is shown in Figure 9.

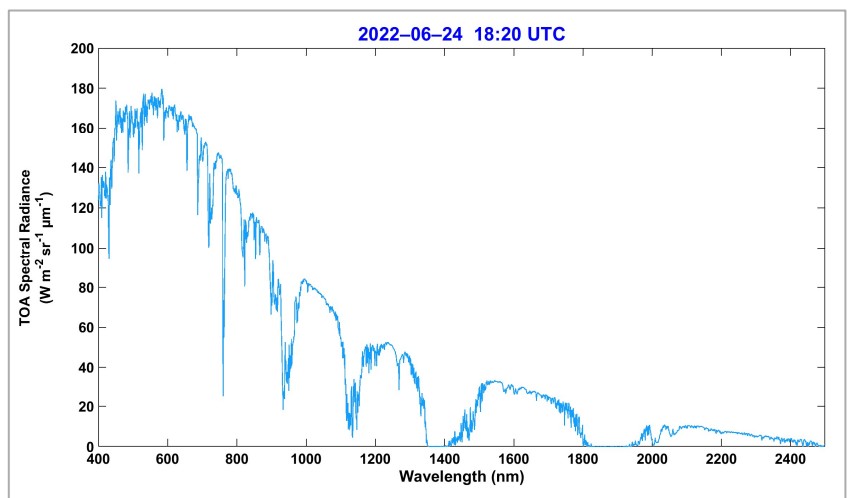

**Figure 8.** TOA spectral radiance at RadCaTS on 24 June 2022 at 18:20 UTC for the Landsat 9 OLI viewing conditions.

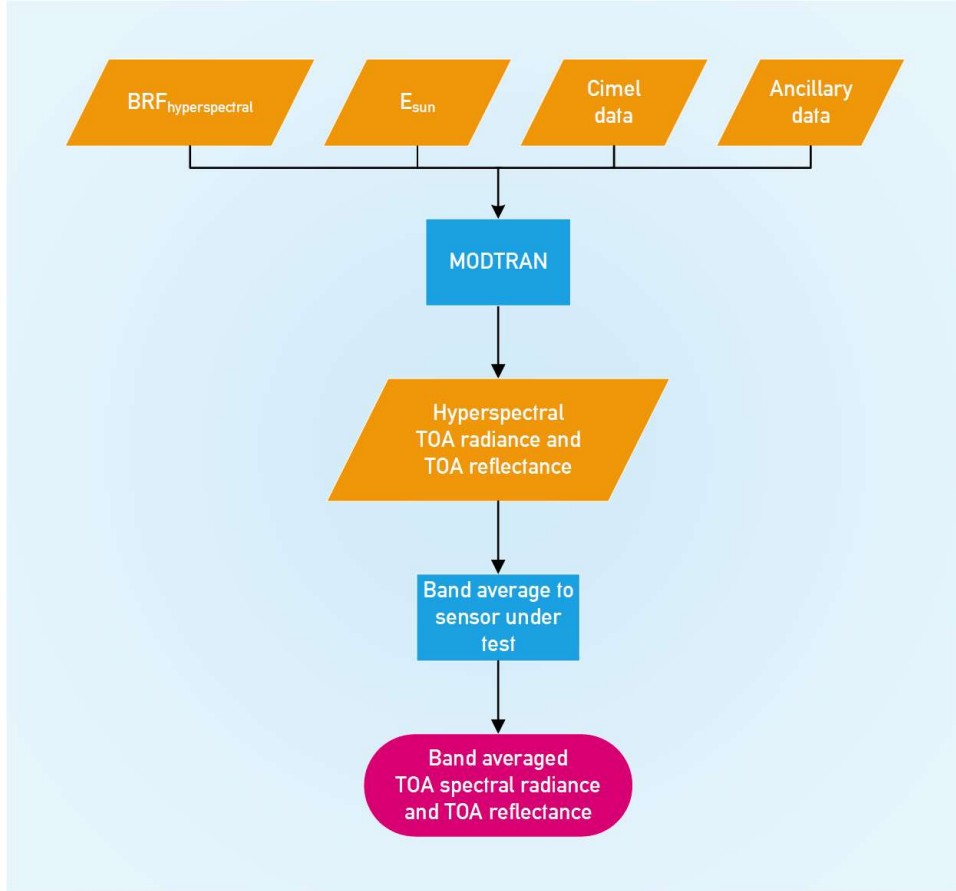

**Figure 9.** RadCaTS TOA spectral radiance (TOA reflectance) processing.

### 2.3. On-Site Personnel: The Reflectance-Based Approach

#### 2.3.1. Overview

The reflectance-based approach to vicarious radiometric calibration is a well-understood technique that has been used successfully over the past 35 years for airborne and space-borne sensors [12–14,18,81,88,94–97]. Like the RadCaTS process, measurements of the atmosphere and surface BRF are made during an overpass time of interest, and the results are used in a radiative transfer code to determine the TOA spectral radiance (or reflectance). The results are then compared to the imagery for the sensor under test.

#### 2.3.2. Field Test Sites

The test site used by UArizona and the NASA GSFC during the brief tandem flight phase of Landsat 8 and Landsat 9 is Ivanpah Playa, California, USA, which continues to be used by a variety of national and international research groups for post-launch radiometric calibration work. As with Railroad Valley, temporal in situ data collection over the past 35 years have allowed UArizona to create a temporal library of hyperspectral surface reflectance data for Ivanpah Playa. It is located southwest of Las Vegas, Nevada, on the Nevada–California border, and it has a useable area of approximately 2 km × 10 km at an altitude of 800 m (Figure 10a). Traditionally, the area east of the Interstate-15 highway has been used for data collection, but this work uses the area to the west of the highway, which was chosen primarily due to easier accessibility.

South Dakota State University (SDSU) has historically used vegetated sites located near Brookings, South Dakota, USA (Figure 10b). The three sites used for this work are 150 m × 250 m, 120 m × 180 m, and 120 m × 180 m in size, at an average altitude of 500 m. The grass at each site is routinely maintained throughout the field campaign season, which typically ranges from April to October.

The reflectance-based approach is the predecessor to the RadCaTS, and the in situ measurements required to obtain TOA quantities follow the same in principle as follows: measurements of the surface BRF and the atmospheric transmission during overpass are required as inputs to a radiative transfer code to determine the TOA quantities of interest, which are then compared to the airborne or spaceborne sensor under test. Thorough descriptions of the atmospheric and surface BRF measurements have been previously published [12,86,94,95,98], but a general description relevant to the Landsat 9 work presented here is included.

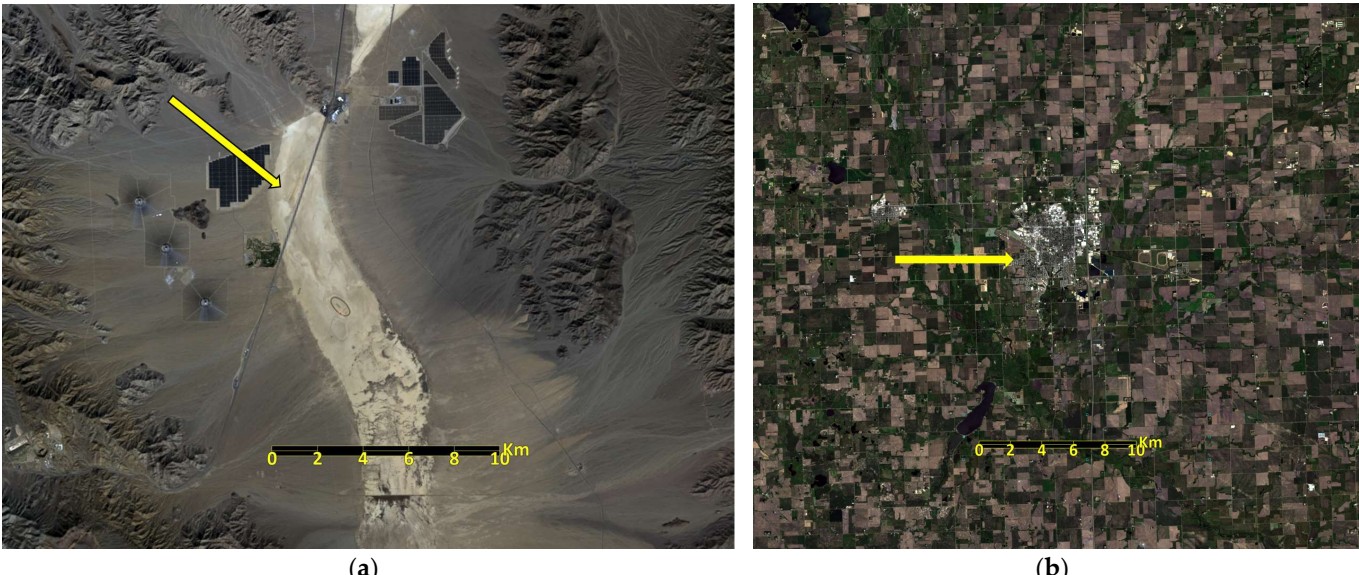

(**a**)          (**b**)

**Figure 10.** (**a**) Landsat 9 OLI image of Ivanpah Playa on 13 November 2021, 18:19 UTC (WRS-2 path 40, row 35); (**b**) Landsat 9 OLI image of Brookings, South Dakota on 27 June 2022, 17:11 UTC (WRS-2 path 29, row 29). Each true color image was created using OLI bands 4, 3, and 2. The yellow arrows in each image denote the approximate location of the ground sites.

2.3.3. Atmospheric Measurements

The atmospheric characterization required during a vicarious calibration field campaign has traditionally been performed using a suite of custom-built and commercial instruments that temporarily operate onsite throughout the time of interest. The goal of such measurements is to determine the atmospheric transmittance, which is used in a radiative transfer code to determine the TOA spectral radiance for a specific time, date, as well as illumination and view geometry.

The research groups in this work make base measurements of solar irradiance extinction due to absorption and scattering with a ten-channel multispectral automated solar radiometer (ASR) that uses narrow bandwidth interference filters for spectral selection. The ten spectral channels are centered at 380 nm, 400 nm, 441 nm, 520 nm, 611 nm, 670 nm, 780 nm, 870 nm, 940 nm, and 1030 nm [26]. The ASR tracks the sun and measures the incoming solar irradiance extinction caused by atmospheric absorption and scattering. The data are then used to derive the total atmospheric spectral optical depths, which can be further broken down into subcomponents such as molecular, aerosol, ozone, and water vapor optical depths [99,100]. In desert sites used by UArizona, the aerosols are modeled by a power-law distribution, and the Angstrom exponent is used to define the aerosol size distribution. Columnar water vapor is determined through a modified Langley approach [27,28], while the columnar ozone amount is obtained from the OMI or OMPS. Ancillary data such as temperature and pressure are measured using additional equipment brought to the site.

Following a manual setup procedure that includes alignment with the sun, the ASR tracks the sun throughout the day through a quadrant cell that is coregistered to the optical axis of the radiometer. Measurements are typically made every minute during the surface BRF data collection process, which is described in the next section. The spectral optical depths are determined using a retrieval based on the Langley method [101]. The optical depth results are used in an inversion method to determine the ozone optical depth and an aerosol size distribution parameter [100]. As mentioned, the work presented here assumes that the aerosol size distribution can be defined by a power law. The main advantage of this assumption is that it only requires one value, known as the Angstrom turbidity parameter, to define the aerosol size distribution.

In the final step of ASR data processing, the optical depths are determined from 350 nm to 2500 nm for use as inputs to the radiative transfer code. The columnar water vapor amount is derived from the ASR data using a modified Langley approach [102]. The Angstrom exponent determined using ASR measurements is also used to compute the Mie scattering phase functions, which are also used in the radiative transfer code.

SDSU uses ASR measurements in conjunction with Langley analyses to derive instantaneous optical depth values. The initial model is propagated from the sun through the atmosphere to ground level, predicting spectral radiance at the reflectance standard, spectral radiance at the grass target, and the diffuse-to-global irradiance ratio of the sky. The predicted results are compared to the radiance measurements made with the portable spectroradiometer, while the diffuse-to-global ratio is compared to a Yankee Environmental Systems shadow-band radiometer to identify any anomalies in the model predictions. The model is then updated and rerun until a fit to reality is obtained.

### 2.3.4. Surface Reflectance Measurements

The surface BRF is typically measured using a portable hyperspectral spectroradiometer having a wavelength range of 350 nm to 2500 nm, and it is carried across the site in a suitable predetermined pattern. The foreoptic is mounted to a boom arm to ensure shadow-free measurements. Periodic measurements of a Spectralon reference panel are made at predetermined points throughout the site collection, and the output digital numbers (DNs) over the two targets (Spectralon and ground) are ratioed to determine the hyperspectral surface BRF for each measurement. Routine measurements of the UArizona and SDSU Spectralon panel BRFs as a function of wavelength and angle are made at the UArizona goniometric laboratory to monitor changes caused by regular field use, including ultraviolet (UV) light and dust exposure [89–91,103]. In the final BRF processing step, the individual transect measurements are averaged together to create one hyperspectral surface BRF to be used in the radiative transfer code for the given sensor overpass.

In addition to the typical hyperspectral spectroradiometer used to collect surface reflectance data, a recently developed multispectral instrument was deployed by GSFC personnel during the November 2021 field campaign. It is known as the Calibration Test Site SI-Traceable Transfer Radiometer (CaTSSITTR), and it was developed by UArizona after the RadCaTS became operational and with the increasing involvement of UArizona in RadCalNet. The main goal in the development of the CaTSSITTR is to allow one-person operation in the field while maintaining an SNR and absolute accuracy on par with laboratory transfer radiometers. There are currently two identical CaTSSITTR instruments as follows: CaTSSITTR-A (Arizona) and CaTSSITTR-G (Goddard) [104–107], and their design specifications mirror those of a RadCaTS GVR, minus the SWIR channel at 1550 nm. The CaTSSITTR measurements at Ivanpah Playa were modeled on the reflectance-based approach, but instead of walking the site taking continuous measurements between panel measurements, the instrument was moved to eight randomly predetermined locations within the overall test site. Operation of a portable spectroradiometer and the CaTSSITTR at Ivanpah Playa and in S. Dakota is shown in Figure 11.

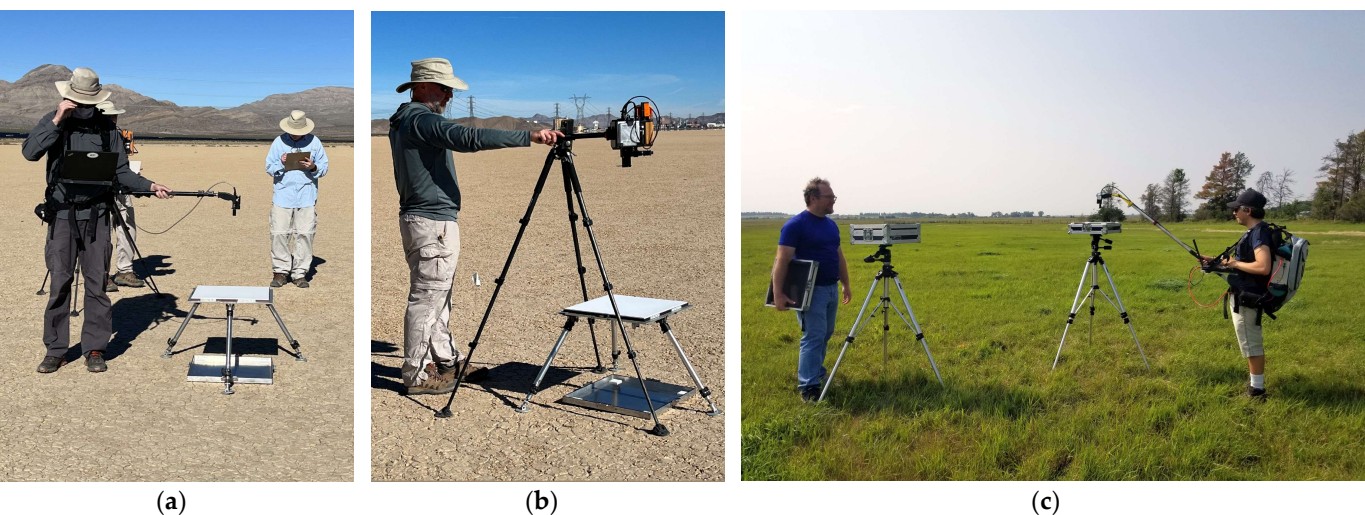

|  |  |  |
|:---:|:---:|:---:|
| (**a**) | (**b**) | (**c**) |

**Figure 11.** Surface BRF measurements. (**a**) UArizona personnel operating a portable spectroradiometer; (**b**) NASA GSFC personnel operating the CaTSSITTR; (**c**) SDSU personnel operating a portable spectroradiometer at a vegetated site. Ivanpah photos courtesy of L. Thome.

### 2.3.5. TOA Spectral Radiance Determination

The final data processing step in the reflectance-based approach is the determination of the TOA quantities of interest using the radiative transfer code [31]. The spatially averaged surface BRF data are used with the atmospheric measurements as input into the radiative transfer code, and the result is the band-averaged TOA spectral radiance and reflectance, which are then compared to the Landsat 9 OLI data products. An example of the TOA spectral radiance determined at one of the SDSU vegetated sites for a Landsat 9 OLI collection on 18 August 2023 is shown in Figure 12. A general summary of the reflectance-based approach processing methodology is shown in Figure 13.

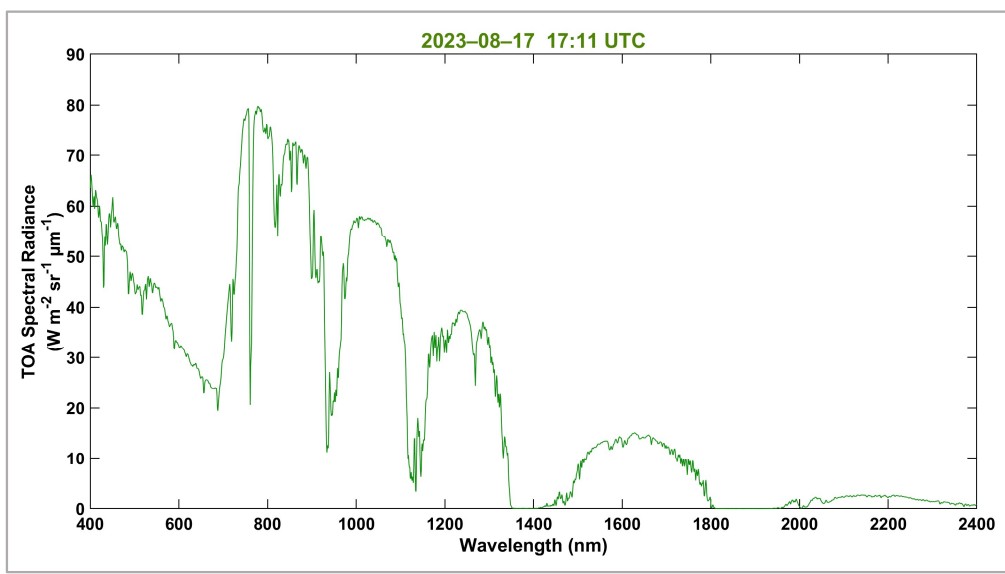

**Figure 12.** TOA spectral radiance at an SDSU vegetated site on 18 August 2023 at 17:11 UTC for the Landsat 9 OLI viewing conditions.

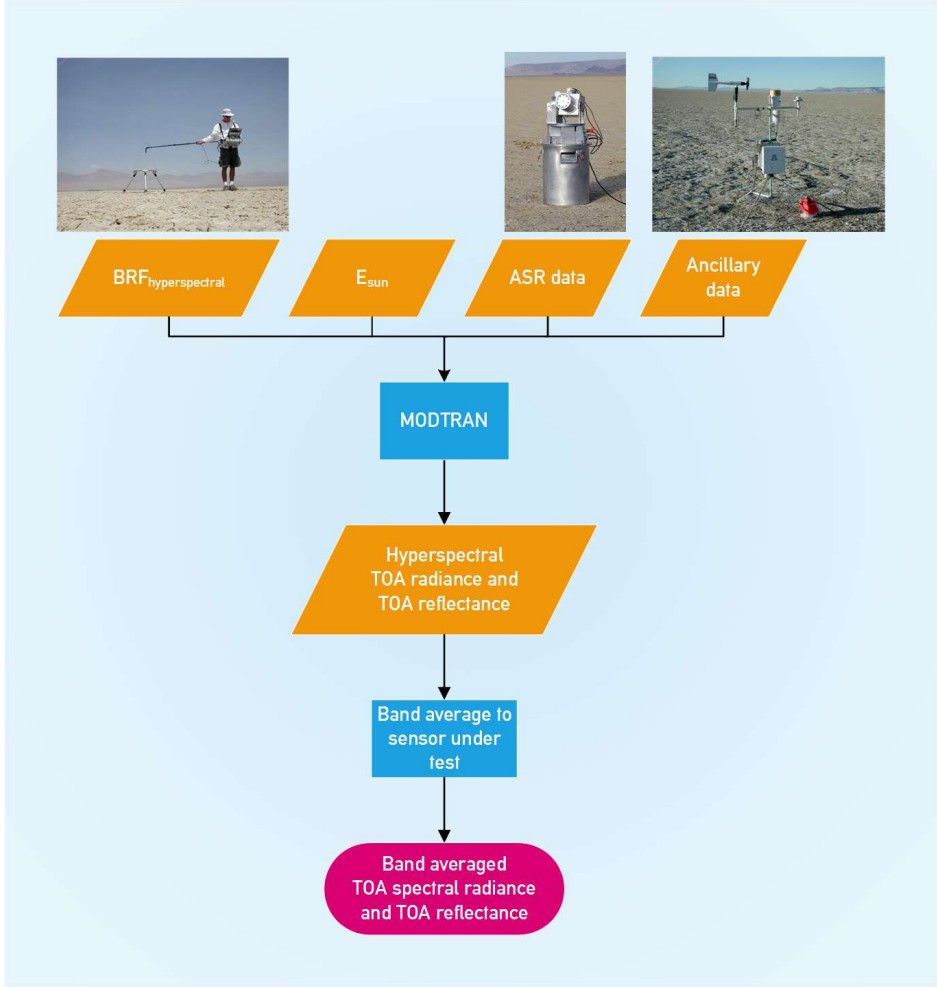

**Figure 13.** A summary of the reflectance-based processing methodology.

## 3. Data

### 3.1. Landsat 9 OLI Imagery

This work uses the Landsat Collection 2 Level-1TP (L1TP) data for comparison with the ground measurements to evaluate the absolute radiometric calibration of the OLI sensor. The L1TP imagery is produced by the Landsat Product Generation System (LPGS), where each image is radiometrically calibrated and orthorectified using ground control points (GCPs) and digital elevation models (DEMs). The imagery is freely available to registered users on the USGS Earth Explorer web portal. As with the Landsat 8 OLI imagery, there are effectively two data radiometry products of interest in each L1TP image as follows: the TOA spectral radiance and the TOA reflectance. The Landsat data archive was restructured into collections in 2016 to ensure that the radiometry and geometry of the Level-1 data products are consistent over the entire data record [108]. Collection 1 included Landsat 1–5, 7, and 8, and it was available until December 2022. It has been superseded with the release of Collection 2, which adds Landsat 9 to the data archive. Updates to Collection 2 include radiometric and geometric improvements, as well as the inclusion of Level-2 surface reflectance and surface temperature data products [109]. The Collection 2 L1TP imagery is converted from a digital number (DN) to either TOA spectral radiance or TOA reflectance using radiometric scaling coefficients that are included in the metadata file accompanying the imagery. It should be noted that the TOA spectral radiance and TOA reflectance products follow separate radiometric calibration traceability chains, so unlike previous Landsat sensors, there is no official exoatmospheric solar irradiance model to convert from one to the other.

Preliminary L1 imagery was made available to the ground calibration teams after the Landsat 9 commissioning phase to monitor any post-launch radiometric issues with the OLI. The calibration parameter files (CPFs) have since been updated, and the work presented here uses the most current CPFs available. As mentioned, the USGS also produces a Level-2 surface reflectance data product (L2 SR) using the Land Surface Reflectance Code (LaSRC), but the scope of this work is limited to radiometric validation.

### 3.2. RadCaTS

A summary of the satellite ephemeris for Landsat 9 at Railroad Valley is shown in Table 2, which describes the collection area of the ground site, the typical overpass time, and the satellite view geometry. There were 38 Landsat 9 daytime overpasses at the RadCaTS during the period from November 2021 to June 2023, with 12 being considered successful. The typical reasons for an unsuccessful collection are clouds at the site or water/snow on the test site surface. The current quality assurance (QA) criteria for an overpass at the RadCaTS is similar to what was developed for the RadCalNet [110]. The QA criteria are founded on a well-understood uncertainty budget and include the 550 nm AOD as well as the average surface reflectance for each of the eight GVR channels to account for surface nonuniformities.

**Table 2.** A summary of the typical overpass time, satellite view zenith angle (VZA), and view azimuth angle (VAA) conditions for Landsat 9 at the RadCaTS. Note that the VAA defined here is measured from the ground site to the satellite.

| RadCaTS (Railroad Valley, NV, USA) | |
| --- | --- |
| Group | UArizona |
| Number of Collects | 12 |
| Collection Area (m) | 1000 × 1000 |
| Time (UTC) | 18:21 |
| VZA (degrees) | 0.5 |
| VAA (degrees) | 103.0 |

### 3.3. Reflectance-Based Approach (SDSU, UArizona, and GSFC)

A summary of the satellite ephemeris for the Landsat 9 OLI at the SDSU sites near Brookings and the joint UArizona–GSFC field campaign at Ivanpah Playa is shown in Table 3. It should be noted that the UArizona and GSFC work at Ivanpah occurred during the tandem flight period of Landsat 8 and Landsat 9, and data are not collected on a routine basis at Ivanpah following this field campaign. The Ivanpah collection was considered successful, so the results are included in the overall analysis. SDSU had three successful Landsat 9 collects in June, July, and August 2022, and they are included in the overall results as well. The reason for an unsuccessful collection at SDSU is typically due to poor atmospheric conditions and water or snow on the site.

**Table 3.** An overview of the overpass time, satellite view zenith angle (VZA), and view azimuth angle (VAA) conditions for Landsat 9 at the sites used by SDSU, UArizona, and GSFC. The VAA is defined as measured from the ground site to the satellite.

| Ground Site | Brookings, SD, USA | Ivanpah, CA, USA |
| --- | --- | --- |
| Group | SDSU | UArizona & GSFC |
| Number of Collects | 3 | 1 each |
| Collection Area(s) (m) | 150 × 250<br>120 × 180<br>120 × 180 | 120 × 300 |
| Time (UTC) | 17:11 | 18:20 |
| VZA (degrees) | 0.6 | 3.8 |
| VAA (degrees) | 104.2 | 283.5 |

## 4. Results

### 4.1. RadCaTS Results

There were 12 successful collects at RadCaTS in the 19 months following the commissioning phase of Landsat 9. The comparison between the OLI and RadCaTS for both the TOA spectral radiance and TOA reflectance data products is shown in Figure 14 as the ratio of the OLI values to those determined using the RadCaTS. There was a higher amount of snow and precipitation than normal at Railroad Valley from January 2023 to April 2023, which is why there is a lower number of successful collects than would be anticipated. Efflorescent salts typically become present while the playa is drying, which can lead to surface uniformity issues when using point measurements, such as those with a finite number of GVRs.

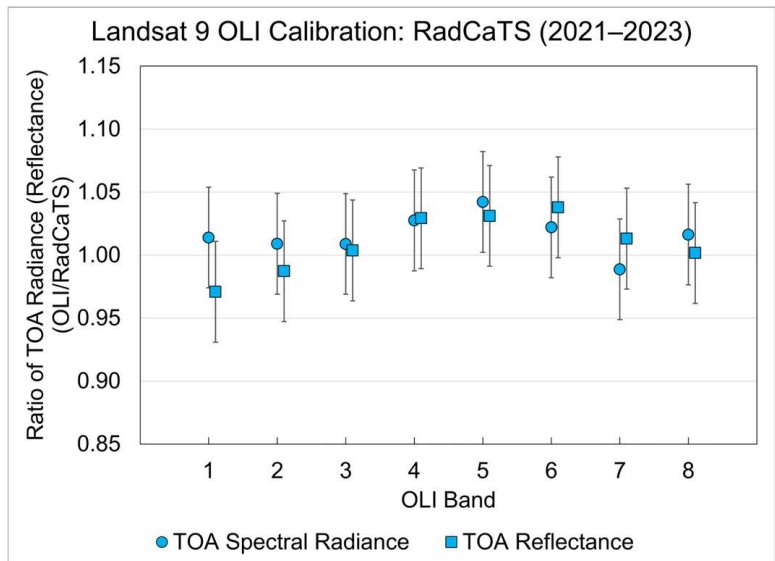

**Figure 14.** A summary of the Landsat 9 OLI radiometric calibration results using 12 comparisons with the RadCaTS (November 2021 to June 2023). The results are shown as the ratio of the TOA quantities for each data product (e.g., spectral radiance or reflectance) to those determined using the RadCaTS.

The RadCaTS results for each of the two data products agree to within the uncertainties of the methodology, which is summarized here as ±4% (k = 1). One noticeable result in Figure 14 is that of the slight offset between the results of the two L1TP imagery products. This bias is highest in the Coastal Aerosol band (band 1) and the SWIR 2 band (band 7), which are those most affected by atmospheric effects and the low SNR of the ground data. This pattern has also been observed in previous Landsat 8 work, but it should be noted that the offset is within the combined uncertainties of the methodologies.

### 4.2. Reflectance-Based Results at SDSU

There were three successful ground-based collects for Landsat 9 during the period from November 2021 to June 2023 using the SDSU vegetated sites in Brookings. The average values are shown in Figure 15, where the format is identical to Figure 14. The uncertainty bars are based on the current uncertainty estimate for the reflectance-based approach of ±2.5% (k = 1) [17]. In OLI bands 1 and 2 (coastal aerosol and blue), there is a higher bias between the OLI and the field measurements, and this is mainly due to the very low surface reflectance of vegetation in these two bands coupled with atmospheric effects that occur for a surface reflectance of <0.2 in these spectral regions. An example of the typical surface BRF as a function of wavelength for Railroad Valley, Ivanpah Playa, and an SDSU vegetated site is shown in Figure 16.

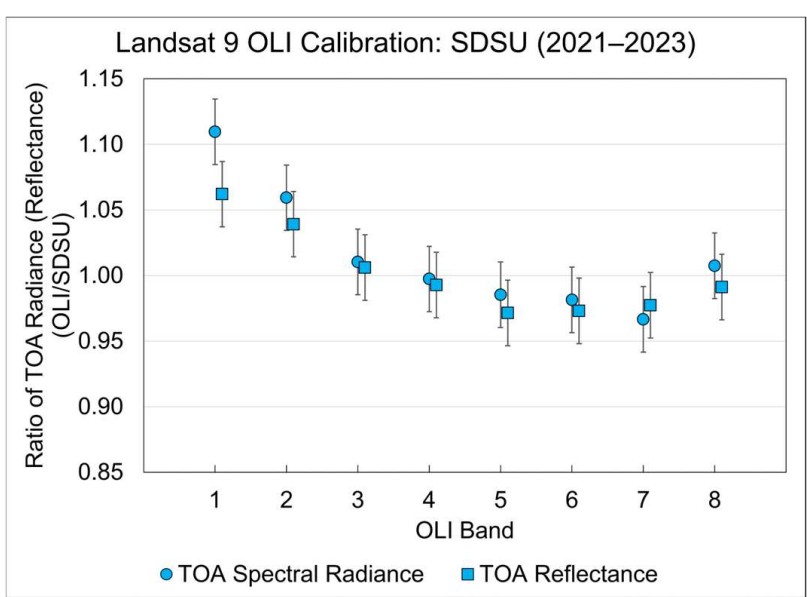

**Figure 15.** A summary of the SDSU radiometric calibration results for the Landsat 9 OLI using three comparisons with ground measurements using the reflectance-based approach (June 2022 to August 2022). The results are shown as the ratio of the TOA quantities for each data product (e.g., spectral radiance or reflectance) to the in situ field results.

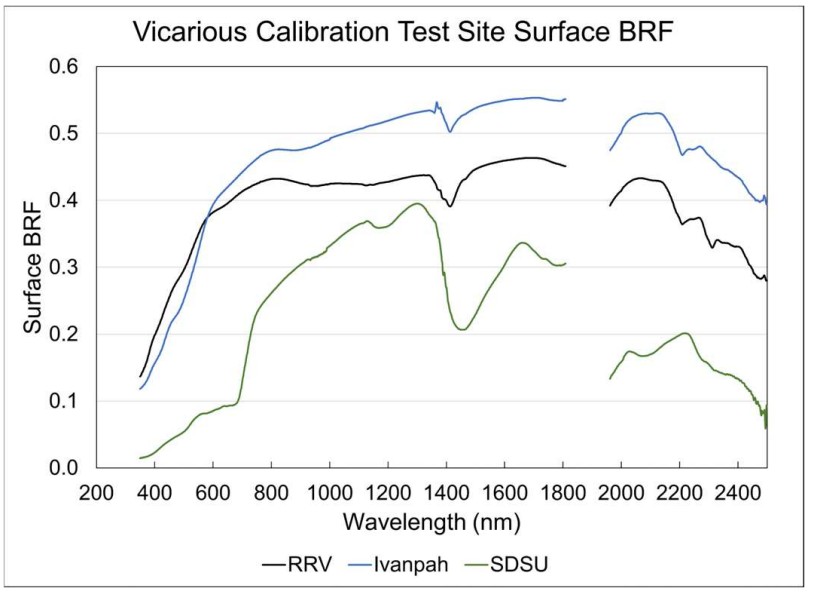

**Figure 16.** An example of the typical surface BRF at Railroad Valley (RRV), Ivanpah Playa, and S. Dakota State University (SDSU) as a function of wavelength.

### 4.3. Reflectance-Based Results for UArizona and NASA GSFC at Ivanpah Playa

The joint UArizona–GSFC field deployment to Ivanpah Playa was successful, with clear skies and minimal wind, which can lead to airborne dust. Each team collected data within the predefined ROI using a portable spectroradiometer (UArizona) and CaTSSITTR (GSFC). Landsat 9 OLI results from the joint UArizona–GSFC collection at Ivanpah Playa during the tandem flight phase with Landsat 8 are shown in Figure 17. The uncertainty bars reflect the current uncertainty estimate for the reflectance-based approach of ±2.5% (k = 1). The results from both teams agree to within the uncertainties of the reflectance-based approach.

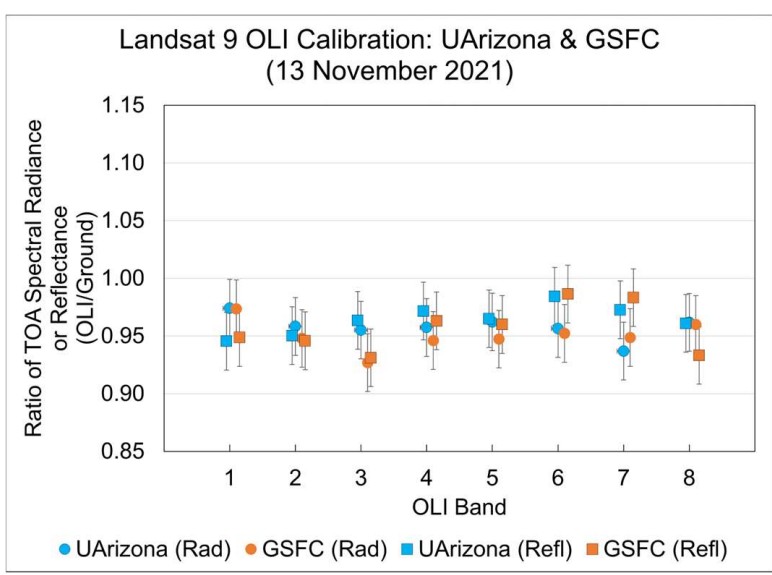

**Figure 17.** A summary of the radiometric calibration results for the Landsat 9 OLI from two ground teams at Ivanpah Playa during the tandem flight phase with Landsat 8 (13 November 2021). The results are shown as the ratio of the TOA quantities for each data product (e.g., spectral radiance or reflectance) to those determined using the reflectance-based approach.

### 4.4. Summary of Combined Landsat 9 OLI Results

A summary of the Landsat 9 OLI results using all data from the three field teams is shown in Figure 18 for the period from November 2021 to June 2023. The results comprise a total of 17 ground-based collects using the RadCaTS and the reflectance-based approach, and the uncertainty bars in this case are the standard deviation (k = 1) of the field results. The results show excellent agreement with both OLI TOA products and are well within the specified radiometric uncertainties of the OLI.

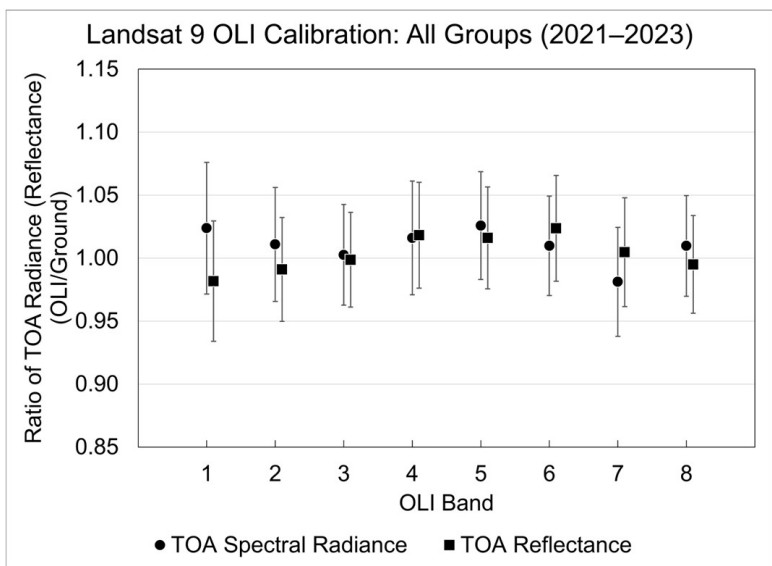

**Figure 18.** A summary of the radiometric calibration results for the Landsat 9 OLI using all available ground-based data, which include the RadCaTS and the reflectance-based results from three groups (November 2021–June 2023). The results are shown as the ratio of the TOA quantities for each data product (e.g., spectral radiance or reflectance) to the ground-based data, and the uncertainty bars for each band are the standard deviation (k = 1) of the field measurements. The same results in tabular form are presented in Table 4.

A summary of the temporal results for TOA spectral radiance is shown in Figure 19, which includes all data from each field team. The results do not show a statistically significant trend, but any such potential trend will become apparent once more temporal data are acquired.

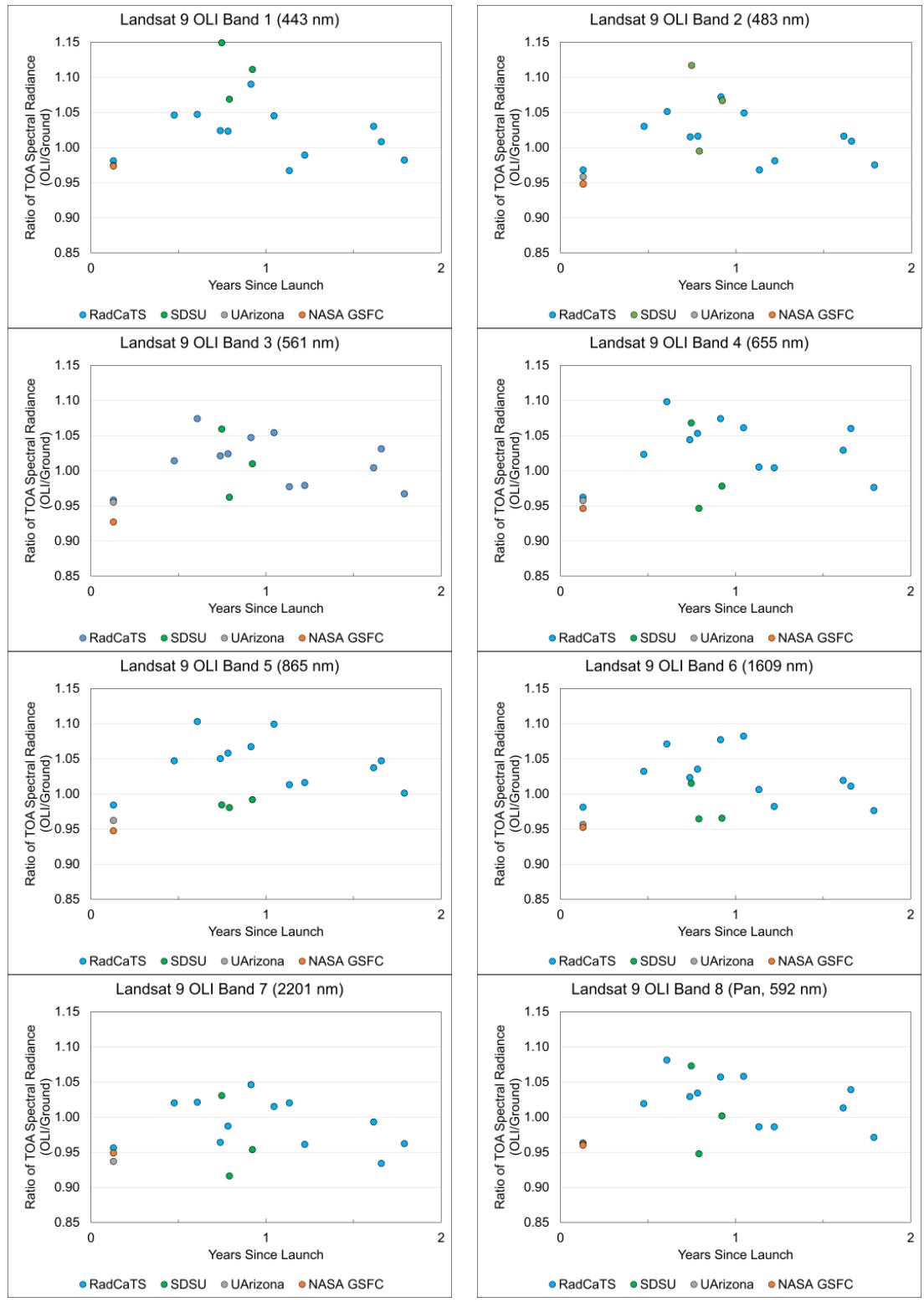

**Figure 19.** The TOA spectral radiance temporal results for the Landsat 9 OLI, comprising all data from each ground team during the period from November 2021 to June 2023.

**Table 4.** The radiometric calibration results for Landsat 9 OLI using all ground data (from November 2021 to June 2023).

| Ratio of TOA Quantities to Ground Measurements—All Field Data | | |
| --- | --- | --- |
| OLI Band (Center Wavelength) | TOA Spectral Radiance (OLI/ground) | TOA Reflectance (OLI/ground) |
| 1. (443 nm) | $1.031 \pm 0.051$ | $0.989 \pm 0.046$ |
| 2. (483 nm) | $1.014 \pm 0.046$ | $0.994 \pm 0.041$ |
| 3. (561 nm) | $1.003 \pm 0.043$ | $0.999 \pm 0.040$ |
| 4. (655 nm) | $1.018 \pm 0.046$ | $1.020 \pm 0.045$ |
| 5. (865 nm) | $1.024 \pm 0.045$ | $1.014 \pm 0.043$ |
| 6. (1609 nm) | $1.008 \pm 0.043$ | $1.021 \pm 0.045$ |
| 7. (2201 nm) | $0.979 \pm 0.040$ | $1.002 \pm 0.040$ |
| 8. (pan, 592 nm) | $1.011 \pm 0.043$ | $0.995 \pm 0.041$ |

One interesting result from the tandem flight phase with Landsat 8 in November 2021 is observed in both the in situ data collected at Ivanpah by two separate ground teams (Figure 17) and at Railroad Valley using the RadCaTS (Figure 19). The Landsat 9 near-nadir overpass at each of these sites on 13 November 2021 is separated by 45 s, and the data from all three collects show similar results in the ratio of OLI TOA values to the ground measurements. This result is encouraging, as the two systems have a different uncertainty traceability path, and they produce results within the combined uncertainty of each methodology. The results from this single day have a smaller ratio of TOA spectral radiance (OLI/ground) than most of the other dates in this work, but the fact that it occurs at two separate sites with separate traceability paths does increase confidence in the result. The results of this work do not show a significant trend in the radiometric calibration of Landsat 9 OLI, but new data are being routinely collected by the RadCaTS and SDSU for future analysis.

## 5. Uncertainty Analysis

A detailed uncertainty analysis of both the reflectance-based approach and the RadCaTS has been previously documented, and only a summary is presented here [17,110,111]. The uncertainty analysis for the RadCaTS was completed as part of the requirements for inclusion as an official RadCalNet site, and the uncertainty of the reflectance-based approach was most recently analyzed for work supporting the launch of Landsat 8. The current non-spectral estimate of the RadCaTS uncertainty is approximately $\pm5\%$ from 400 nm to 2400 nm, and the estimate for the reflectance-based approach is approximately $\pm2.6\%$ in the middle of the visible spectral region.

## 6. Conclusions

After an extensive ground-based field measurement campaign by various government and university groups during the first nineteen months of operation, it has been confirmed that the Landsat 9 OLI instrument is performing exceptionally well and has maintained its stability throughout this period. The ground-based data were compared to the L1TP imagery available from the USGS through Earth Explorer. The results showed that, on average, the OLI instrument aligns with the design specifications for TOA spectral radiance ($\pm5\%$) and TOA spectral reflectance ($\pm3\%$) in all multispectral bands as well as the panchromatic band (band 8).

The temporal analysis did not reveal any significant degradation pattern in the OLI instrument over time. Furthermore, the results obtained from the automated RadCaTS facility at Railroad Valley supported the findings of the ground personnel, with even slightly better agreement with the OLI than the in situ measurements conducted by on-site personnel. However, it is worth noting that the number of RadCaTS data sets was lower than expected throughout the year, primarily due to the unusually high frequency of poor weather conditions experienced during that period.

The radiometric calibration of the Landsat 9 OLI will continue to be measured and monitored throughout its operational life, employing reliable and established techniques that are currently in use by a wide array of calibration teams. These techniques include ground-based in situ measurements using both automated systems and on-site personnel, observations over invariant desert sites, lunar observations, and cross-comparison with other well-calibrated sensors. By utilizing these methods, the temporal accuracy and precision of Landsat 9 OLI data will remain well understood, thereby ensuring the continued production of high-quality and reliable Earth observation data for a wide range of applications.

**Author Contributions:** All the authors were involved in various aspects of the data collection and/or analysis during the period of this work. J.S.C.-M. and N.J.A. were involved in the in situ collection and analysis of field data, which included simultaneous field work with K.J.T. and B.N.W., who provided the NASA GSFC results. C.T.P. and L.M.L. were involved in on-site data collection at the SDSU calibration site, including supervision, and they also assumed responsibility for the subsequent processing and analysis of field data. All authors have read and agreed to the published version of the manuscript.

**Funding:** The UArizona work was supported by the USGS cooperative agreement G20AC00003, as well as NASA grants 80NSSC21K1749 and 80NSSC20K1872. The SDSU work was supported by USGS EROS grant G18AS00001.

**Data Availability Statement:** Data from the RadCaTS are freely available to registered users on the Radiometric Calibration Network (RadCalNet) web portal: www.radcalnet.org (accessed on 1 January 2024). Landsat 8 and 9 OLI Collection 2 data are courtesy of the US Geological Survey DOI Object Identifier (DOI) number: /10.5066/P975CC9B.

**Acknowledgments:** The authors would like the thank the U.S. Bureau of Land Management (BLM), Tonopah, Nevada, office for assistance and permission for the use of Railroad Valley and the BLM, Needles, California, office for their assistance in using Ivanpah Playa. We would also like to thank NASA AERONET for processing the Cimel sun photometer data, the NOAA Global Monitoring Laboratory (GML) for the $CO_2$ data, and the OMI and OMPS teams for the ozone data.

**Conflicts of Interest:** The authors declare no conflict of interest.

**Appendix A**

The surface BRF, $\rho$, for each of the GVR channels is determined using the following equation:

$$\rho = \frac{\pi \, C_{GVR} \, (V_{GVR} - V_{GVR\,dark})}{\frac{E_0}{d^2} \tau_A \cos\theta + E_{sky}}$$

where $C_{GVR}$ is the radiometric calibration coefficient of the GVR (W m$^{-2}$ sr$^{-1}$ μm$^{-1}$) V$^{-1}$, $V_{GVR}$ is the output voltage (V), $V_{GVR\,dark}$ is the dark voltage (V), $E_0$ is the exoatmospheric spectral solar irradiance when the Earth–Sun distance is one astronomical unit (AU) (W m$^{-2}$ μm$^{-1}$), $d$ is the Earth–Sun distance (AU), $\tau_A$ is the direct solar beam transmission (unitless), $\theta$ is the solar zenith angle, and $E_{sky}$ is the diffuse spectral sky irradiance (W m$^{-2}$ μm$^{-1}$).

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
