# Peer review of "The Ground-Based Absolute Radiometric Calibration of the Landsat 9 Operational Land Imager"

_remotesensing, doi:10.3390/rs16061101_

Round 1

Reviewer 1 Report (Previous Reviewer 1)

Comments and Suggestions for Authors

I confirmed the revised paper. I have no more comments.

Author Response

Thank you for taking the time to review this paper.

Reviewer 2 Report (Previous Reviewer 2)

Comments and Suggestions for Authors

The added information addresses most of my concerns and improves the manuscript.

There are only a few minor changes that are needed, mostly where useful new information was included in the review response but did not also appear in the manuscript.

From my first review:

"In order for these methods to work, you need to know something about the atmospheric scattering, particularly for the shortest wavelength bands. (In fact, I feel that the statement at line 112 is wrong: you need a measurement of the surface reflectance, atmospheric transmittance, *and* something about the atmospheric scatterers.)"

The authors responded:

"We are assuming that the atmospheric spectral transmittance includes multiple scattering, so the two terms were grouped together as part of ‘transmittance’."

But, the MODTRAN runs include aerosol scattering models, and the ground measurements either include aerosol property retrievals from AERONET, or use a retrieval method from the ASR that splits the gas absorption (ozone) from the molecular and aerosol scattering. So I still don't see how this statement makes sense. Perhaps if the scattering was isotropic you could just effectively ignore the separate contributions of absorption and scattering to the extinction, but that's not what your methods appear to do. I still think this sentence (now at line ~120) needs to be corrected.

From initial review:

"Also, at this same stage (inputs to RT code) - what other trace gases are included, and are they just assumed to be at a standard atmosphere concentration? (e.g., methane?)"

Authors' response:

"The trace gases include NH3, NO, NO2, SO2 and HNO3. Water vapor, the uniformly mixed gases, ozone, and the trace gases together make up the 12 MODTRAN default band model species."

This should be stated as well in the manuscript, (probably section 2.2.5 in the revision). I assume these other trace molecules are used with standard atmosphere concentration profiles? Please state what assumption was made about the concentrations as well and include that in the manuscript.

From initial review:

"Finally, in the results section (4), separate comparison is made for the OLI reflectance and radiance. I am very confused about why the two OLI quantities have different spectral biases relative to the ground observations. The only explanation that comes to mind is that the OLI

L1 algorithm assumes a different top-of-atmosphere solar irradiance than the authors are using in their analysis. Aren't both quantities (OLI rad vs radiance, or ground-based rad vs radiance) only different by their assumed solar spectral irradiance? If that's true, then it seems like the analysis should be done with a matching solar spectral irradiance. Some explanation needs to be added here."

Authors' response:

"Landsat 8 and 9 OLI do not use an exoatmospheric solar irradiance model, unlike Landsat 7 ETM+ and earlier platforms originally did. The TOA spectral radiance product is based on the preflight calibration, and the TOA reflectance product’s traceability chain is based on the onboard solar diffusers. The two techniques have a bias, and this is observed by various ground teams."

Ok, I think this makes sense, in that the two quantities are effectively derived from different calibration methods on-orbit, and thus have different traceability chains. I would add a few sentences somewhere describing this fact. Maybe in section 3.1 when you describe the Landsat data you could mention the two products and why they are analyzed separately.

From initial review:

"Line 347: since the "CPFs" are changing with time, there needs to be extra information added here to unambiguously describe which parameters were used in the analysis. Ideally there should be a data product version number that would identify this. If not, then perhaps a data download data would be sufficient?"

Authors' response:

"When extracting data from the Landsat imagery, it is the metadata file that is included with the L1 and L2 imagery that is used to convert the imagery DNs to TOA quantities. The metadata file is updated using the most recent CPFs. I’m unclear as to what ‘...a data download data...’ is, but perhaps the reviewer means a ‘data download date’? If so, that has been updated."

Yes that was just a typo, I meant "date". I don't see that a data download date was added anywhere. However, I now see there is a formal DOI from USGS in "Data Availability Statement" at the end, which I must have missed before. Assuming that DOI ties the analysis to a specific "CPF" version number, that is all that is required.

Author Response

Thank you for taking the time to review this paper.

Reviewer 3 Report (Previous Reviewer 4)

Comments and Suggestions for Authors

The paper is reasonably comprehensive. I have only a few minor comments for the authors to consider.

1.     For sky spectral radiance measurement, what direction does the Cimel detector aim at?

2.     What is the uncertainty of the estimated solar irradiance on the test site surface from the

Chance-Kurucz (ChKur) model?

3.     In Fig. 5, can the results from the GVRs be plotted in different colors? With a colored plot, a reader can better sense the uncertainties associated with the GVR measurements.

4.     Fig. 7 substantially helps me understand the procedure in determining the test site surface BRF. One minor point is to clearly show that if the difference in the two BRFs are too large, then the new feed-in BRF is the new hyperspectral BRF (to be further refined).

5.     I wish the paper provides an Appendix that mathematically shows the relationship between the test site surface BRF and Esky.

6.     Line 448: Can the authors elaborate a bit on diffuse-to-global irradiance ratio of the sky?

7.     ASD appears in the text a few times, but I do not see the definition.

Author Response

Thank you for taking the time to review this paper.

This manuscript is a resubmission of an earlier submission. The following is a list of the peer review reports and author responses from that submission.

Round 1

Reviewer 1 Report

Comments and Suggestions for Authors

I think this paper is appropriate to be published. This type of paper describing the accuracy evaluation of L1 radiance (reflectance) is very important for the product users. The following comments may be considered for betterment.

1) Band width and spectral shape description of each band may be beneficial for readers to understand its methodology.

2) Reflectance range difference between sites is also beneficial for readers. Currently, detailed RadCaTS site BRF and SDSU site reflectance (in LINE-405, <0.2 in blue bands) are only described.

3) It may be better to describe a difference of the methodology and limitation between VNIR and SWIR bands and within SWIR bands. For example, the comparison between SWIR-1 (1.6 micron) and SWIR-2 (2.2 micron) difference, such as water vapor effects and SNR limitation, is interesting for readers. In current version only SWIR-2 (2.2 micron) SNR difficulty is described in LINE-393. 

4) RadCaTS measurement in LINE-162 is up to 1550nm, which does not cover 2.2 micron. Do you calculate 2.2 micron reflectance from 1.6 micron reflectance?

5) The title of section 3.3 includes “cross-calibration with Landsat 8 OLI”. However, I cannot find the result of cross-calibration results is this paper. Many users are interested in consistency between L8 and L9, I think.

Reviewer 2 Report

Comments and Suggestions for Authors

Review of "The Ground-Based Absolute Radiometric Calibration of Landsat 9 OLI"

This manuscript covers the results of the first ~ 2 years of vicarious radiometric calibration data collections for the Landsat 9 Operational Land Imager (OLI). The authors briefly describe the vicarious calibration methodologies and the three different ground sites used. The results show that the ratios between the OLI data and the ground calibration measurements are within the design specification of Landsat 9 OLI, indicating good on-orbit calibration. These results are an important independent assessment of the absolute radiometric calibration of the OLI. Such measurements and analysis are a critical foundation for satellite datasets.

Overall, this is a very useful contribution to the literature to document the performance of the Landsat 9 OLI in the early mission period. It will serve as a key reference and piece of documentation to connect the analyses to the topic of vicarious calibration more widely. The manuscript is also clearly a good match to the journal topics and the special issue in particular.

The methodology is sound, and the authors have been at the forefront of developing these techniques over the years and successfully applying them to a wide variety of satellite sensors. However, there are a number of areas where the description is unclear, where additional explanation or missing details need to be added. In some cases the figures also should be improved. I stress these changes would substantially improve the clarity of the paper, particularly for readers that are not as directly involved in Landsat or these vicarious calibration activities.

Major issues:

I found the description of the two methods difficult to follow, I think because the key differences are mentioned in multiple places scattered throughout sections 2 and 3. It would help to have a flow chart showing the key inputs for each method, and highlighting what steps involve measurements vs assumed datasets. In addition, a summary table about the instrumentation differences would be useful. For example, the fact that RadCATS has an AERONET vs the solar radiometer at other sites is an important distinction.

In order for these methods to work, you need to know something about the atmospheric scattering, particularly for the shortest wavelength bands. (In fact, I feel that the statement at line 112 is wrong: you need a measurement of the surface reflectance, atmospheric transmittance, *and* something about the atmospheric scatterers.)

There is a lack of detail in addressing how scattering is handled. My guess is either the data is screened for the lowest possible AODs (so simplifying assumptions about the aerosols are less impactful), and/or perhaps the AERONET data are giving aerosol single scattering albedo (SSA) and particle size that can be used in the Radiative Transfer (RT) model. E.g.: at lines 197 - 202, there must be some input to the RT code about the aerosol SSA & particle size, not just the AOD, correct? Is the SSA/particle size just assumed? The angstrom exponent could be a proxy for particle size - but at line 199 it indicates it is only used to translate AOD from 500 to 550 nm.

How does this process work with the ASR approach at the other sites? The description of the ASR suggests that aerosol SSA cannot be estimated from the data.

Also, at this same stage (inputs to RT code) - what other trace gases are included, and are they just assumed to be at a standard atmosphere concentration? (e.g., methane?)

On the "reflectance BRF": no where is this acronym defined, and there are important subtleties that may arise depending on what specific quantity is used. I assume this to be the Bi-directional Reflectance Factor (specifically: the ratio between the observed reflectance and a perfect Lambertian reflector, at a specific observation geometry). The manuscript does not discuss the angular dependence of the BRF anywhere - I assume this is ignored because the ground locations are taken at the same time as the Landsat observations and approximately same observation angle since OLI is at near-nadir. However this might be difficult for the second method, since the ground based instruments need to be carried across the site - if this takes ~ hours, then doesn't the sun angle change enough to have an effect? The authors should add a few sentences somewhere to make these assumptions more explicit.

Finally, in the results section (4), separate comparison is done for the OLI reflectance and radiance. I am very confused about why the two OLI quantities have different spectral biases relative to the ground observations. The only explanation that comes to mind is that the OLI L1 algorithm assumes a different top-of-atmosphere solar irradiance than the authors are using in their analysis. Aren't both quantities (OLI rad vs radiance, or ground-based rad vs radiance) only different by their assumed solar spectral irradiance? If that's true, then it seems like the analysis should be done with a matching solar spectral irradiance. Some explanation needs to be added here.

Minor issues:

There are numerous formatting issues throughout that need to be fixed ("Error! Not a valid bookmark self-reference" appears in the review draft.)

Line 58: OLI has 9 bands, but then you only discuss 8 - I assume this is because the 9th band (1.38 um) cannot be calibrated with these methods; please add a sentence or two to clarify.

Line 308 ' aerosols are modeled by a power law distribution': what specific aerosol property is assumed to be a power law?

Line 347: since the "CPFs" are changing with time, there needs to be extra information added here to unambiguously describe which parameters were used in the analysis. Ideally there should be a data product version number that would identify this. If not, then perhaps a data download data would be sufficient?

Suggestions:

Table 1: I would list the OLI band FWHMs here as well.

Figure 6: it would be very helpful to add a similar TOA spectral radiance plot for the SDSU site. It would support the claims made at line ~402 about why the SDSU results look different.

Figure 11: the plot symbols are switched in a way that makes this visually hard to compare to earlier figures. In Fig 11, symbol shape denotes Rad vs Refl, while color denotes the ground team. In Fig 9, 10, both symbol and shape denote Rad vs Refl. I would use either shape OR color to denote Rad vs Refl in figure 9, 10, and then use the additional property in Figure 11 to denote ground team.

Conclusions section: while I agree that there is no temporal trend in the radiance ratios, the patterns do appear quite spectrally correlated (for example the plots for Band 1 and 2 are very similar). It would be interesting to know what process drives this spectral correlation: presumably this is related to limitations in aerosol characterization which leads to biases in the RT modeling of the TOA radiance. If the authors have any ideas about possible sources of residual bias (that would drive the spectrally correlated error) that would be of interest to readers.

Reviewer 3 Report

Comments and Suggestions for Authors

This manuscript makes a significant contribution to understanding the radiometric calibration of Landsat 9 OLI-2.  The overall structure of the paper and the clarity of the writing are outstanding and make for a very readable paper. I would like to complement the authors for taking the time to present their work with such care.  I've made a few comments - all are completely up to the authors discretion.  

Reviewer 4 Report

Comments and Suggestions for Authors

The paper presents an important work that establishes ground calibration sites and uses the sites to calibrate LandSat-9 Operational Land Imager. Although the frame of the topics in the paper is very good, the paper is quite descriptive. As a result, I am unable to understand the paper quantitatively. I’d like to suggest the following for the authors to make the paper more comprehensive.

1.     How many (7?) GVRs are deployed to calibrate Railroad valley Test Site, and what are the locations of the GVRs deployed? Would like see a figure to show the GVR locations (and the GVR index).

2.     Because the BRF of a test site depends on the both the incident and outgoing directions, is the 10-degree viewing angular range of the GVR too large to make the BRF measurement inaccurate? I’d like to see the paper discusses this point.

3.     It seems that the GVR only looks downwards, presumably in parallel to the test site surface normal. However, the paper does not describe how a GVR orientation is established. Simply by eye?

4.     I’d like to see a figure to show the BRFs from all GVRs, one point for each GVR, and also along the time dimension. Figure 3 has no time dimension. In Figure 3, which point is for which GVR?

5.     Because the OLI is a pushbroom sensor, to cover a large swath on Earth, I expect OLI’s each band consists of a very large array of detectors. Can the authors specify this a bit: How many detectors for each band? The detectors in a band do not look at the Earth with the same direction. A 185 km swath on Earth at the 705 km height above the ground means that the angular span of the detectors is very large: roughly 185/705*57.3 deg = 15 deg. I’d like to see the paper explains the BRF versus the viewing angle: Is the BRF flat enough across the viewing angles of the detectors? Are there any measurements and/or references?

6.     I’d like to see some description on the Cimel sun photometer. Is it true that the Cimel photometer can directly measure the solar irradiance and the sky irradiance? The two irradiances differ by a large factor. What is the viewing angular range of the Cimel photometer? In addition, for the sky measurement, which direction does the Cimel aim at? (Test Site to satellite direction?)

7.     Can the authors describe a bit on why the solar irradiance and the sky irradiance are enough to figure out the atmospheric quantities?

8.     Figure 9. The paper claims that the reflectance based measurement leads to a smaller uncertainty than the radiance based approach. Why in the figure, the error bars from the two approaches are identical?

9.     Figure 12. How are the uncertainties calculated? What are the main uncertainty contributors and how large for each contributor?

10.  How do the authors perform band-average: A simple average?

11.  Figure 13. What are the respective averages for OLI/ground? There are many points for each band, but the average (across the time and test sites) is what a reader may finally get on the performance of the OLI. In addition, there are many detectors for each OLI band, are the detectors perform similarly?
